# The Influence of Short Video Platform Characteristics on Users' Willingness to Share Marketing Information: Based on the SOR Model

Rui Shi , Minghao Wang, Chang Liu * and Nida Gull

School of Economics and Management, Yanshan University, Qinhuangdao 066004, China
* Correspondence: lchaoren318@126.com

**Abstract:** User marketing information sharing plays a significant role in boosting the effectiveness of short video marketing. Exploring the factors influencing the willingness to share has become a meaningful way to improve the dissemination of marketing information. This study examines how the characteristics of short video platforms affect users' willingness to share marketing information and investigates the mediating role of users' flow experience. We adopted qualitative and quantitative analyses to explore the issue. Twelve participants were recruited to conduct the online interviews and 306 valid data points on users' willingness to share marketing information were obtained through a questionnaire. The study was conducted using structural equation modelling (SEM) and mediating effect tests. The interviews verified that the information quality, the service quality, and the system quality are essential characteristics of a well-established short video platform. The results of the empirical analysis showed that the information and the service and system quality of short video platforms have a positive impact on users' willingness to share marketing information through perceived control and pleasure, respectively. The effect of information quality on users' desire to share marketing information is the strongest. This study provides a reference for short video platforms to optimize and improve their marketing effectiveness.

**Keywords:** short video platforms; marketing information sharing; flow experience; information systems success; stimulus-organism-response

## 1. Introduction

The 21st-century online network is increasing worldwide, and relevant data has shown that the global online video user base is expected to reach 3 billion by 2023 [1]. The short video platform has become an indispensable social platform in the internet era by its colossal user advantages [2]. The content of short videos has expanded from a pure entertainment perspective into areas such as news dissemination and learning science [3]. There is a new form of selling short videos live with goods, bringing short videos more deeply into users' lives. In the based-on-short-video consumer industry boom, short video platforms have become a new channel for more and more companies to distribute their products [4]. Companies have entered the short video platform to attract users with high-quality short videos due to their user market and business value. The platform's commenting, retweeting, and private messaging features increase user engagement and have a significant impact on users' purchase intentions [5]. This new marketing model has put the short video platform on track for commercial realization. In this case, there is also a significant upward trend in the proportion of users making consumer decisions [6].

As people's living standards improve, so does the consumer demand for personalized goods. The short video sales model is more intuitive, breaks the time and space boundaries of consumption, and has better interactivity [7]. Consumers can place their orders with more comprehensive product information [8]. Leveraging consumer social network relationships can boost business. It is an effective measure to increase the effectiveness of

marketing, which spread marketing messages quickly through consumers' social circles [9]. Consumers are the channel through which information is exchanged. The more marketing information consumers share, the more significant the impact will be [10]. Public people in social networks or people with a high level of intimacy take the initiative to share information after they have knowledge and understanding of the product. It will lead to the further expansion of marketing information [11]. This gives the marketing process a fractured tendency.

With the rapid growth of the short video industry, the research topic of willingness to share marketing information has been a focus of academic discussion. On the level of the consumers themselves, Li et al. (2020) mentioned that user apprehension plays an important role in users' willingness to share information [12]. User social skills are also an important influencing factor. Jiang et al. (2020) found that the higher the influence level of the sharer in their circle of friends, the stronger the willingness of users to share marketing information [13]. Fischer et al. (2018) investigated the influence of users' neurological factors on information-sharing behavior by recording the amplitude of electroencephalographic (EEG) oscillations while subjects were watching short videos [14]. In terms of the consumption environment, information content and attributes are important influencing factors. The presentation and environment of marketing information stimulate users' emotions and influence their subsequent sharing and forwarding behavior [15]. Guo et al. (2021) found that the professionalism and trustworthiness of information can directly or indirectly influence users' willingness to share information [16]. Previous scholars have also argued that users' willingness to share marketing information is influenced by two factors. The first is the external motivation of the expected reciprocal relationship and the second is the internal motivation of perceived pleasure [17]. With social media users as the research objectives, Peng et al. (2018) found that the influence between publishers and consumers is beneficial when the content is novel. However, the impact will be reduced if the content is widely distributed [18]. From the existing research findings, most studies in the literature have investigated the topic of willingness to share marketing information from the user perspective and the consumer environment. Few research studies have approached the issue of willingness to share marketing information from the user front-end, which is the perspective of platform characteristics. Platform characteristics are important factors that directly show users the comprehensiveness and high-quality content of the platform and then determine the marketing structure. It has greatly affected the experience of consumers in the consumption process and has practical significance for commodity marketing. Thus, this study considers the platform characteristics and selects the user flow experience as a mediator to explore the marketing information-sharing mechanism.

Taking the above arguments together, this study's main contributions are as follows. Firstly, we explored the issue of users' willingness to share marketing information from a platform perspective. This complements past research on user perspectives and provides theoretical support for analyzing influencing factors. Secondly, the aim of this study was to obtain objective data through qualitative and quantitative research methods, which assure accurate model validation. Thirdly, this study explored the pathways that influence users' willingness to share marketing information. In addition, it provides some guidance and lessons for short video platforms and creators in developing marketing plans at a practical level. Fourthly, exploring the factors that influence the willingness to share marketing information has an important impact on improving the quality of short video output and marketing effectiveness.

## 2. Theoretical Background and Hypothesis Development

### 2.1. SOR Theory

The SOR theory, also known as the stimulus–organism–response theory, explores the role of various stimuli on an individual's cognitive or psychological responses and the subsequent generation of behavioral responses [19]. The theory points out that all aspects of the environment will play a vital role, affecting the person's internal state and then forming

the response behavior [20]. The generation of the information-sharing willingness of short video marketing users is a typical dynamic process of external information stimulation causing changes in the organism and generating information-sharing willingness [21].

The SOR theory is widely used in related research on consumer behavior and provides an essential theoretical basis for studying the formation path and influencing factors of user behavior [7]. Han et al. (2021) used the SOR theory to study the impact of shelf displays on consumer purchasing behavior [22]. The SOR model explores the relationship among social media interactivity, perceived value, an immersive experience, and continued purchase intentions [23]. The SOR theory is used in this paper to build a clear theoretical framework by considering the characteristics of short video marketing platforms as the stimulus (S), the use and satisfaction experience of short video marketing users as the organism (O), and the willingness of users to share marketing information as the response (R). In doing so, it explores how the external environment stimulates the experience of short video users and further generates a willingness to share marketing information.

*2.2. Short Video Platform Characteristics as Stimulus*

DeLone and McLean (2003) proposed the information systems success theory in 1992. The original model was initially used to assess information systems and was later modified based on the utility of the successful aspects of e-commerce systems. The updated information system success model is collectively called the D&M model [24]. The six dimensions of the updated model are information quality, system quality, service quality, usage, user satisfaction, and net income. The newly added service quality allows for a better overall assessment of information system quality in conjunction with system quality and information quality. Abdulrazaq et al. (2019) examined the usefulness of trust on the use and success of e-government services based on information systems success theory. They found that people's trust in the government is directly influenced by the information quality, the service quality and their actual use [25]. Shim et al. (2020) examined how health information websites' information quality, system quality, and service quality affect user satisfaction and perceived benefits. The results showed that information quality contributes to user satisfaction and perceived benefits. There is a significant positive relationship between service quality and user satisfaction, while the role of system quality is not empirically supported [26]. Ma's (2021) study on the factors influencing live customer happiness discovered that the information quality, the services provided, and the system's efficiency greatly impacted satisfaction [27]. Fang et al. (2021) studied the factors influencing university students' willingness to use digital university libraries. The questionnaire data showed that the system quality of digital libraries significantly influenced university students' satisfaction [28]. The application of information system success theory provides a complete reference for the study of information systems. The information quality, service quality, and system quality of short video platforms significantly impact the user experience and the consumption process. Therefore, this study focuses on these three dimensions of willingness to share information.

*2.3. Perceived Control and Perceived Pleasure as Organism*

The flow experience has been defined as a feeling of being fully immersed in an activity [29]. There are two main views among researchers on the dimension of the flow experience. The first is the study of the flow experience as a single dimension. Based on the SOR model, Yin et al. (2021) built a model of the influence mechanism of an online shopping platform AI marketing technology experience on consumers' purchase intention. The results showed a significant role of flow experience between insight and purchase intention [30]. Cuevas et al. (2021) applied flow experience theory to the social search process in social media to verify the impact of content quality and system quality on consumer purchase intentions [31]. The other is as a multidimensional study. Zaman et al. (2010) stated that flow experience can be measured by two factors: perceived pleasure and perceived control [32]. Zhang et al. (2021) believed that the three dimensions of emotional

satisfaction, potential sense of control, and loss of self-awareness could better explain the flow experience [33]. Since users' willingness to share information is mainly based on the fluency of platform system operation and the pleasure brought by information, this paper takes perceived control and perceived pleasure as the measurement dimensions of flow experience.

*2.4. The Impact of Short Video Platform Characteristics on User Experience*

Perceived control is the level of control an individual has over their environment and actions. Information quality is the evaluation of information provided by an information system, mainly referring to the output quality of product content on short video platforms, including the integrity of the marketing process and the comprehensiveness of product information that merchants provide. Users analyze and organize the information internally according to its content before reacting to the stimuli they receive. Suppose short video platforms and merchants update their information content promptly to match users' demands. In that case, it will increase the level of control users have over their information. After accessing the information, the user will analyze the current information utilization environment according to the perception control mechanism. This is combined with the information quality to make the appropriate forwarding behavior [34]. At the same time, high-quality information allows users to experience a higher level of information matching. This reduces their information search efforts and further enhances their sense of control over the information and the platform [35]. Schwarz et al. (2020) found that effective information can increase the interaction between individuals and information, further enhancing an individual's sense of control over the environment in which they live [36]. Service quality is one of the key factors in becoming a marketing success and refers to the level and quality of service provided by the platform operator. Personalized service is particularly prominent in the context of e-commerce marketing and has become an important prerequisite for enhancing user satisfaction. A high quality of service gives the user better control over the consumption process and adequate control allows the user to feel a higher quality of service [37]. Li et al. (2022) discussed the role of perceived control in mediating the relationship between service staff and customer comfort [38]. System quality is interpreted as a measure of the performance of a platform's information system itself. The system quality on a short video platform is mainly manifested in the platform's functionality, stability, and operability, which directly affect the user's experience. The fluency of the user's use process will enhance the user's control ability and willingness to operate actively. Conversely, a system that lacks relevance and contains disorganized pages will reduce the user's interest in browsing, distracting them from the system and thus reducing their sense of control over its use [39]. Abdallah et al. (2016) studied learning management systems and found that system quality can improve teachers' sense of control, thereby improving the teaching quality [40]. Xu et al. (2022) found that the system quality of short video platforms had a significant effect on the control of user actions and could promote users' willingness to search for information [41]. As such, we hypothesized:

**Hypothesis 1 (H1).** *The information quality of short video platforms positively affects consumers' perceived control.*

**Hypothesis 2 (H2).** *The service quality of short video platforms positively affects consumers' perceived control.*

**Hypothesis 3 (H3).** *The system quality of short video platforms positively affects the consumer's perceived control.*

Perceived pleasure refers to the user's state of physical and mental pleasure in the whole process, emphasizing personal feelings and stimulating people's internal motivation to act. Information quality is an essential factor affecting user attitudes. High information quality can significantly improve the pleasure of the user's consumption process [27]. The

richness, vividness, and reliability of the information on a website positively impact user pleasure [42]. Meng et al. (2022) studied the user experience of a mobile visual search. The study found that the integrity of information had a positive impact on information quality and affected user satisfaction through perceived usefulness [43]. The service quality is the information interaction between the short video platform, the merchant, and the user, which requires the user's doubts and problems to be solved quickly. At the same time, it covers the individual demands of users and the user's perception of the services provided by the platform during the whole process [44]. High quality platform service quality can increase user visits, thus further promote user behavior. [45]. Previous studies have found that perceived quality with service content positively affects perceived pleasure [46]. The quality of service as perceived by users affects their satisfaction and pleasure. A high level of system quality makes the user experience more pleasant and generates a pleasant feeling of use [47]. Papakostas et al. (2022) studied AR-simulation training systems and they found that smooth systems can enhance user experience and behavioral intentions [48]. However, there is a lack of research on the effect of the systematic quality of short video platforms on users' willingness to share marketing information. As such, we hypothesized:

**Hypothesis 4 (H4).** *The information quality of short video platforms positively affects consumers' perceived pleasure.*

**Hypothesis 5 (H5).** *The service quality of short video platforms positively affects consumers' perceived pleasure.*

**Hypothesis 6 (H6).** *The system quality of short video platforms positively affects consumers' perceived pleasure.*

*2.5. The Impact of User Perception on Willingness to Share Marketing Information*

The willingness to share information as an individual's dynamic behavior is closely related to the user's flow experience. It has been shown that the duration and intensity of a user's flow experience has an impact on their willingness to share. In the research on the information-sharing behavior of social networking sites (SNS), it has been found that user perception control plays a significant role in promoting user information-sharing behavior [49]. Perceived pleasure emphasizes personal feelings and is an intrinsic motivator for action. Users receive information and services in such a way that their expectations are met, which in turn leads to pleasurable emotions. Savolainen (2015) found that positive emotions can stimulate individuals to seek and share information [50]. The success factors of the information system allow people to recognize the operational experience, information, and services given by the platform, creating a sense of control and pleasure. As a result, they will take the initiative to share information. As such, we hypothesized:

**Hypothesis 7 (H7).** *Perceived control positively affects users' willingness to share marketing information.*

**Hypothesis 8 (H8).** *Perceived pleasure positively affects users' willingness to share marketing information.*

*2.6. The Mediating Role of Perceived Control and Perceived Pleasure*

The outcome variables of flow experience are mostly individual attitudes and intentions. In recent years, numerous studies have combined flow experience theory to explore individual responses to external stimuli [51]. Moreover, it has a clear mediating role between external stimuli and individual responses. In a study of online retail shop website design, Kühn et al. (2018) argued that flow experience is a mediating variable for visual appeal and perceived usefulness to influence consumer purchase intentions [52]. During Fan's (2022) study of consumer impulsive buying behavior in live e-commerce, it was found that the flow experience acted as a full-chain mediator in an indirect role [53]. As such, we hypothesized:

**Hypothesis 9a (H9a).** *Perceived control mediates the relationship between the characteristics of the short video platform and the willingness of users to share marketing information.*

**Hypothesis 9b (H9b).** *Perceived pleasure mediates the relationship between the characteristics of the short video platform and users' willingness to share marketing information.*

This study proposes the following research framework by using the SOR theory as the research framework, together with information system success and flow experience. The theoretical framework of this study is presented in Figure 1.

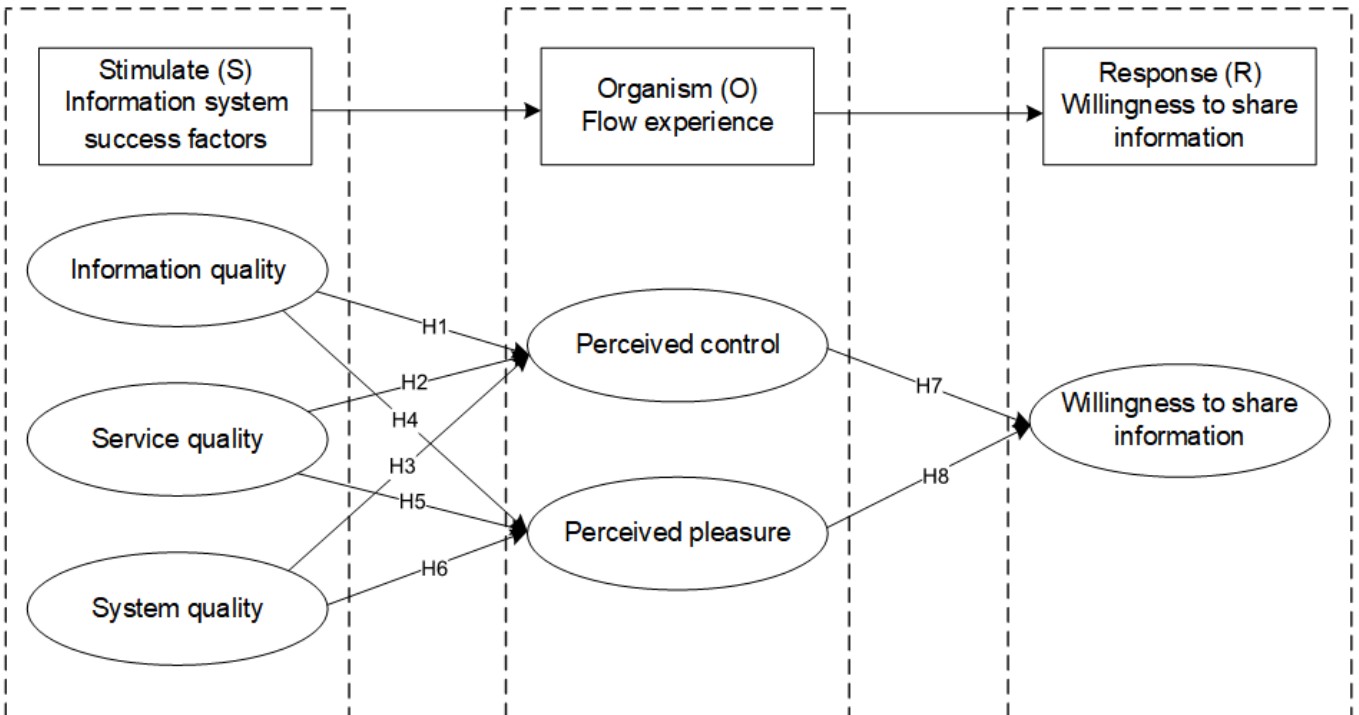

**Figure 1.** Proposed study model.

### 3. Methodology and Data Analysis Results

Qualitative and quantitative methods are not mutually exclusive in research. They are interlinked and complementary. These two research methods have been widely used in social science research in recent years [54]. The study was designed to obtain real data in order to accurately explore the issue of willingness to share marketing information. We have adopted a mixed research approach (a mixed method combining qualitative and quantitative methods) as adopted by Zhang et al. (2022) to further deepen our research ideas [55]. Qualitative methods enable the researcher to conduct a behavioral analysis based directly on the performance of the person being interviewed. The method is to clarify the specific structure of each theory in reality and to have comprehensive control over the overall research structure.. The quantitative approach allows the use of statistical analysis software to test the reliability of hypotheses on actual sample data [56]. The results of the quantitative analysis provide a visual representation of the relationships between the variables.. We conducted online interviews to verify the validity of the hypothetical model based on this qualitative analysis. We then published a questionnaire on the willingness to share to analyze the various paths of the model quantitatively. The combination of the two types of methods ensured the accuracy of the research.

### 3.1. Qualitative Research

3.1.1. Sample and Data Collection

Four postgraduate students, two undergraduate students, three logistics members, and three teachers were selected from the School of Economics and Management at Yanshan University to participate in our interviews. All participants were proficient at operating smartphones and short video platform software and have experience consuming short video platforms. Due to the epidemic's impact, the twelve interviewees agreed to an online interview. Their personal information is shown in Table 1 with an interview time limit of 45−60 min. The specific questions in Appendix A guided this interview.

**Table 1.** Participants' information.

| No | Gender | Age | Education | Profession | Platform Shopping Frequency (Monthly) | Platform Browsing Time (Daily) | Monthly Income (RMB) |
|---|---|---|---|---|---|---|---|
| 1 | Male | 23 | Master's | Student | 4 times | 2−3 h | 1600 |
| 2 | Male | 25 | Master's | Student | 2 times | 1−2 h | 1500 |
| 3 | Female | 23 | Master's | Student | 5 times | 1−2 h | 1800 |
| 4 | Female | 24 | Master's | Student | 3 times | 0−1 h | 1500 |
| 5 | Male | 19 | Undergraduate | Student | 3 times | 1−2 h | 1200 |
| 6 | Female | 18 | Undergraduate | Student | 5 times | 2−3 h | 1500 |
| 7 | Female | 43 | Undergraduate | Logistics | 3 times | 2−3 h | 4500 |
| 8 | Female | 35 | High School | Logistics | 6 times | 2−3 h | 5000 |
| 9 | Male | 47 | Specialist | Logistics | 3 times | 0−1 h | 3800 |
| 10 | Female | 36 | PhD | Teacher | 5 times | 0−1 h | 6000 |
| 11 | Female | 38 | PhD | Teacher | 4 times | 0−1 h | 6000 |
| 12 | Male | 54 | PhD | Teacher | 2 times | 0−1 h | 8000 |

The interviewees responded accordingly with their situation, while the researcher took detailed notes.

3.1.2. Results of Interview

After compiling the results of the interviews, we found that all twelve participants considered quality information, services, and systems to be the characteristics of a well-established short video platform. During the usage of short video platforms, individuals were more willing to engage in active behavior when the information quality, the service quality and system quality met their requirements. The reason is that individuals will perceive the operation as being under their control rather than being accepted passively. Participant 1 stated: "When shopping for products using the short video platform, I like the comprehensive presentation as it will not take up much of my time. Meanwhile, the customer service attitude affects my mood, and the smooth system makes me want to continue to use the platform. When all this meets my needs, I feel dominant and think I am in control of my buying process". Participant 5 stated: "I know the short video platforms can capture my information preferences and I do not feel offended when I receive information which matches my preferences. I hope that the merchant will be able to answer my questions precisely, so that I feel in control of the whole process". The interviews were partially at odds with our initial assumptions regarding perceived pleasure, despite most interviewees considering that quality information, services, and systems enhance the perceived pleasure of individuals. Participant 3 stated: "I feel the information that was pushed met my requirements and the smoothness of the system would have satisfied me, but I don't think the quality of the service would have had a greater impact on me". Participant 9 stated: "I think consumers should receive a high-quality service from both the platform and the merchant. But I would be happy to have access to quality information and operating systems". In terms of individual experience, the interviewees agreed that a sense of control and pleasure would increase their willingness to share marketing information.

Participant 10 stated: "When I can feel in control of the whole and the platform provides information, services and systems that make me feel satisfied thus creating pleasure. I would love to share this marketing information with anyone I think needs it". Participant 12 stated: "I share information only when others need, thus the quality of the information is very important. Additionally, quality services and systems will increase my willingness to share information".

In summary, our model structure setting was in line with reality. Our assumptions remained consistent with the interviewees' thoughts and had some validity. A more scientific approach to the hypotheses was followed by a quantitative approach to test them.

### 3.2. Quantitative Research

### 3.2.1. Sample and Data Collection

The research was mainly for consumers who had shopping experiences on short video platforms. We excluded respondents who did not meet the requirements during the questionnaire collection and analysis process. The first group comprised the people who did not have a shopping experience from short video platforms. The second group comprised the people who had highly consistent questionnaire results or no differentiation. The third group comprised the people who gave contradictory answers or reacted to data falsely. The questionnaires were distributed through the professional questionnaire website Questionnaire Star and Credamo. Among the 331 questionnaires collected over half a month, 306 were valid, with a valid return rate of 92.4%. The descriptive statistics of this questionnaire showed that 44.4% and 55.6% of the respondents were men and women, respectively. Both sexes were fairly evenly represented. By education level, the majority of students were undergraduates and postgraduates. The percentages were 64.9% and 18.3%, respectively. In terms of age, the age groups of 21 to 30 and 31 to 40 years old were predominant. They accounted for 64.2% and 19.0%, respectively.

### 3.2.2. Survey Instrument

Modifications to the variables were made by collating the results of existing research and by referring to established scales from existing studies. The information system success theory refers to Huang (2018) and we added a new "personalized services for users" question to the service quality variables [57]. The flow experience refers to the research results of Koufaris (2002) and the dimensions proposed by it were optimized [58]. Using perceived control and perceived pleasure as the dimensions of flow experience, a questionnaire was designed on the influencing factors of users' willingness to share marketing information.

The questionnaire contained six variables: information quality, service quality, system quality, perceived control, perceived pleasure, and willingness to share information. Each variable was designed with two to three questions, with 17 items. The measurement of variables adopted the Likert five-point scale method, using an integer between 1 and 5 to represent the user's attitude to the problem. Then they were asked to choose according to their actual situation. The questionnaire was pre-surveyed before its official release to ensure the accuracy of the data. In terms of data processing, the scale's reliability and validity were analyzed using SPSS 26.0 statistical software and AMOS 22.0 software. Finally, the AMOS 22.0 software was used to obtain the degree of fit, path coefficient, and total effect index to analyze the model.

### 3.2.3. Data Result Analysis

### Reliability and Validity

The reliability of the data obtained from the questionnaire was tested and analyzed through SPSS 26.0 software. Firstly, we used Cronbach's α coefficient to test the reliability of the potential variables. The results showed that the Cronbach's α of the subscales of information quality, service quality, system quality, perceived control, perceived pleasure, and willingness to share information were 0.839, 0.782, 0.817, 0.7776, 0.840, and 0.789, all

of which were higher than 0.7. Furthermore, the Cronbach's $\alpha$ of the entire sample data reached 0.942, indicating that the design of the scale and the reliability of the obtained data were of high quality and the data had good reliability.

Whether the data were suitable for factor analysis was tested by a KMO test and the Bartlett spherical test. The KMO value in the data was 0.948, indicating that they met the requirements. The Bartlett spherical test statistic was 3032.391, and the degree of freedom was 136. It can be inferred that there was a strong correlation between the variables, which was suitable for the subsequent factor analysis. This was followed by a principal component analysis and a maximum-variance balanced rotation to determine the structure of each element. A total of six common factors were extracted through the factor extraction variance cumulative contribution rate, and the cumulative explained variance was 75.398%.

We constructed a model of the factors influencing users' willingness to share marketing information. Then we used AMOS 22.0 to estimate the model and to explore the fit between the latent and observed variables. According to the output results, it was found that the factor loads of all items in their corresponding dimensions were more significant than the traditional value of 0.5, which met the standard requirements. This showed that the scale had good structural validity in the overall design, and the structural validity passed the test.

The factor loads of information quality, service quality, system quality, perceived control, perceived pleasure, and willingness to share information in the scale were all greater than 0.6, as shown in Table 2. This indicated that the latent variables were highly representative of the topics to which they belonged. In addition, the AVE value of each latent variable was more significant than 0.5, and the composite reliability CR was more significant than 0.7, indicating that the observation indicators had good consistency and that the convergent validity of the data was ideal. Finally, a discriminant validity test of the observed variables was carried out. The results are shown in Table 3. The square root of the AVE value of each variable was significantly larger than the correlation coefficient between the observed variables. This indicated that the scale had good discriminant validity. In conclusion, the data of this questionnaire passed the reliability and validity tests.

**Table 2.** Confirmatory factor analysis results.

| | | Path | Estimate | AVE | CR |
|---|---|---|---|---|---|
| q3 | <— | Information quality | 0.793 | | |
| q2 | <— | Information quality | 0.748 | 0.632 | 0.837 |
| q1 | <— | Information quality | 0.842 | | |
| q6 | <— | Service quality | 0.791 | | |
| q5 | <— | Service quality | 0.680 | 0.552 | 0.787 |
| q4 | <— | Service quality | 0.754 | | |
| q9 | <— | System quality | 0.793 | | |
| q8 | <— | System quality | 0.737 | 0.599 | 0.817 |
| q7 | <— | System quality | 0.790 | | |
| q15 | <— | Perceived pleasure | 0.832 | | |
| q14 | <— | Perceived pleasure | 0.741 | 0.636 | 0.839 |
| q13 | <— | Perceived pleasure | 0.816 | | |
| q10 | <— | Perceived control | 0.814 | | |
| q11 | <— | Perceived control | 0.665 | 0.538 | 0.776 |
| q12 | <— | Perceived control | 0.713 | | |
| q16 | <— | Willingness to share information | 0.811 | 0.653 | 0.791 |
| q17 | <— | Willingness to share information | 0.806 | | |

Structural Model Analysis

In this study, the AMOS 22.0 software was used to construct the structural equation for model fitting. In AMOS, we evaluated structural models using fit criteria from CMIN/DF, RMSEA, CFI, GFI, NFI, and IFI. Most of the criteria exceeded the specified thresholds:

CMIN/DF = 1.709, RMSEA = 0.048, CFI = 0.974, GFI = 0.936, and IFI = 0.974. This showed that the research model had an excellent fitting effect with the data and had good explanatory and predictive power, which confirmed that it was an acceptable model.

**Table 3.** Discriminant validity.

|  | Information Quality | Service Quality | System Quality |
|---|---|---|---|
| Information quality | 0.632 |  |  |
| Service quality | 0.413 | 0.552 |  |
| System quality | 0.384 | 0.379 | 0.599 |
| $\sqrt{AVE}$ | 0.795 | 0.743 | 0.774 |

Path coefficient estimates is shown in Figure 2. The structural equation model results showed that all path coefficients' absolute values ranged from 0 to 1. Except in the case of Hypothesis 5, the *t*-test corresponding to each hypothesis reached a significant level of 0.05. According to the size of each path coefficient, the system quality had a more significant impact on the Perceived control than the information quality and service quality. The impact of information quality on perceived pleasure was more significant than the service quality and system quality. Compared with perceived control, perceived pleasure significantly impacted users' willingness to share information.

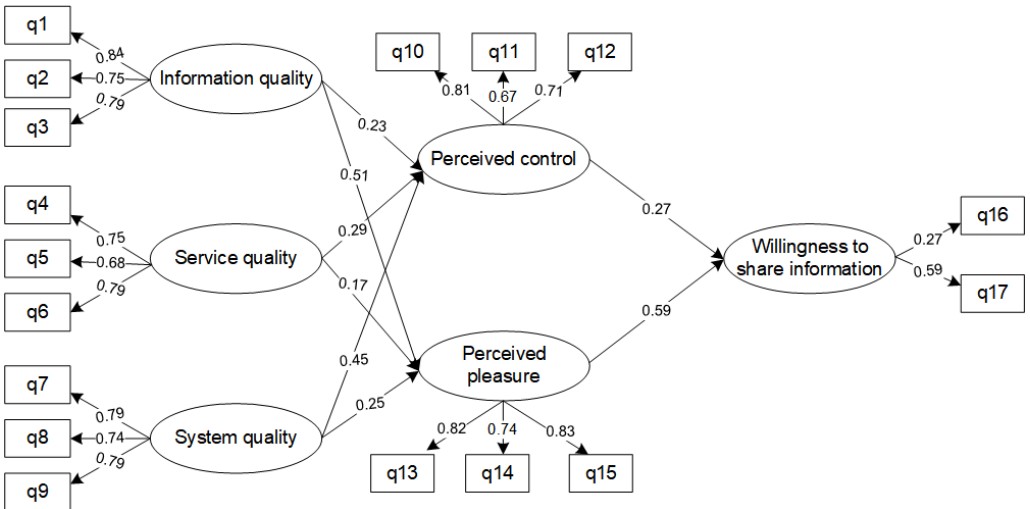

**Figure 2.** Results of structural equation model.

Hypothesis Test

The data found the *p*-values for Hypotheses 1−8 to be within the standard range of 0.05 except in the case of H4. Assuming that the *p*-value of H4 was 0.24, which is greater than the standard value of 0.05, it indicates that the positive effect of service quality on perceived pleasure was not verified, as shown in Table 4. By reviewing and sorting out the literature, it was found that most researchers believe the perceived of pleasure will promote people's willingness to act and produce positive behavior. Su's study found that service quality positively impacts behavior intention through individual satisfaction. However, it has no direct and significant effect on behavior intention, and the data analysis results in this paper are consistent with Su's research results [59]. There are two possible reasons for this situation. On the one hand, users receive and process information on short video platforms, and the services provided by the platforms are what users should enjoy as consumers. This is standard information delivery and not a critical factor in making the platforms more enjoyable for users. On the other hand, users solve their problems according to the services of the merchants and the platform, which only reduces their

doubts about unknown content. The solution to the problem does not bring the user to a level of enjoyment. Therefore, the effect of service quality on the user's perceived pleasure is insignificant.

**Table 4.** Model-fitting results.

| | Path | | | Standardized Estimates | Non-Standardized Estimates | S.E. | C.R. | *p* | Result |
|---|---|---|---|---|---|---|---|---|---|
| H1 | Perceived control | < | Information quality | 0.231 | 0.213 | 0.102 | 2.092 | 0.036 | Yes |
| H2 | Perceived pleasure | < | Information quality | 0.511 | 0.511 | 0.112 | 4.56 | *** | Yes |
| H3 | Perceived control | < | Service quality | 0.293 | 0.294 | 0.148 | 1.985 | 0.047 | Yes |
| H4 | Perceived pleasure | < | Service quality | 0.166 | 0.181 | 0.154 | 1.174 | 0.24 | No |
| H5 | Perceived control | < | System quality | 0.448 | 0.457 | 0.125 | 3.656 | *** | Yes |
| H6 | Perceived pleasure | < | System quality | 0.248 | 0.274 | 0.13 | 2.101 | 0.036 | Yes |
| H7 | Willingness to share | < | Perceived control | 0.267 | 0.274 | 0.116 | 2.356 | 0.018 | Yes |
| H8 | Willingness to share | < | Perceived pleasure | 0.591 | 0.558 | 0.111 | 5.024 | *** | Yes |

Note: *** $p < 0.001$.

Mediation Test

The short video platform brings new experiences and affects users' perceived control and pleasure by its information, services, and systems. To explore the specific relationship between users' willingness to share information and platform characteristics, a model was constructed using perceived control and perceived pleasure as mediating variables based on the framework of the SOR theory. The study used the bootstrap method for the mediating effect analysis. The bootstrap sample size was set to 5000 and the test was performed with a 95% confidence interval for significance. The test results are shown in Table 5. All paths did not include 0 between the upper and lower limits of the 95% confidence level. This shows that the mediating effect was significant, and both perceived control and perceived pleasure played a partly mediating role between short video platform characteristics and users' willingness to share marketing information. Hypotheses 9a and 9b were both tested.

**Table 5.** Results of the mediation effect test.

| Mediation Path | Effect Size | Std. Error | Bootstrap 95% CI | | Result |
|---|---|---|---|---|---|
| | | | LLCI | ULCI | |
| Information quality→Perceived control→Willingness to share | 0.141 | 0.0375 | 0.0727 | 0.2181 | Supported |
| Information quality→Perceived pleasure→Willingness to share | 0.2259 | 0.0394 | 0.1496 | 0.3038 | Supported |
| System quality→Perceived control→Willingness to share | 0.1572 | 0.032 | 0.0963 | 0.222 | Supported |
| Service quality→Perceived pleasure→Willingness to share | 0.2158 | 0.0382 | 0.1417 | 0.2894 | Supported |
| System quality→Perceived control→Willingness to share | 0.1286 | 0.0347 | 0.0666 | 0.2041 | Supported |
| System quality→Perceived pleasure→Willingness to share | 0.1866 | 0.0343 | 0.1206 | 0.2542 | Supported |

## 4. Discussions and Conclusions

This study combined information system success theory and flow experience theory to investigate how platform characteristics affect users' willingness to share marketing information based on the SOR theoretical model. The results of Study 1 identified the characteristics of a well-established short video platform. Study 1 also initially investigated the correlation between the characteristics of short video platforms and users' willingness to share marketing information. The results of Study 2 further revealed that the short video platform's information quality, service quality, and system quality positively impacted the user's willingness to share marketing information. In addition, perceived control and perceived pleasure played an important mediating role between them.

The findings are consistent with previous research in that information quality had the most significant impact on user-perceived pleasure, and system quality had the most significant impact on user-perceived control. A previous study showed that the information quality and the system quality create a sense of pleasure and control for users, leading to a willingness to share [60]. Indeed, the primary consideration for users to share marketing

information is information quality. High-quality information can stimulate the satisfaction of user experience, generate a strong sharing motivation, and reduce the difficulty of user selection so that users can quickly select information recipients. The system quality affects the user's platform control process. The use of a complete platform system can significantly enhance the user's control experience. The whole process of users receiving, browsing, and sharing information will be more in line with platform and merchant expectations.

Both perceived control and perceived pleasure can positively affect users' willingness to share marketing information, and the impact of perceived pleasure on users' willingness to share marketing information is more pronounced. This shows that the user's perceived pleasure in the process of browsing information improves satisfaction and enables users to make positive decisions. The degree of influence of perceived control is weak, and it puts the user in control of his own behavior and environment. In this state, users are more rational and do not make behavioral decisions based on emotional perceptions.

A short video platform's salient features are information quality, service quality, and system quality [24]. There is also an apparent relationship among the three, which jointly affects the user experience and synergistically affects the user's willingness to share marketing information. The correlation between information quality and system quality is significantly greater than other characteristics. The quality of the system directly affects the fluency of users to obtain information, so the system quality can be inferred to be more critical to the overall consumer experience.

From the perspective of users' flow experience, perceived control and perceived pleasure play a mediating role between the characteristics of the short video platform and the user's willingness to share marketing information. The mediating effect of perceived pleasure is stronger than perceived control. The reason may be that most users are willing to share out of their positive emotions, which is in line with Mansourian's (2022) research [61]. Users' positive emotions become the connecting link or a catalyst between the marketing information and the sharing willingness.

## 5. Research Implications

There are some implications of this study. First, the study extends the application of the SOR theory. The three stages of stimulus–organism–response effectively explain the connection between the individual and the outside world. It offers an example of a practical application of SOR theory. Second, the study explores the factors that influence the willingness of users to share short video marketing information through their perceptions, starting from the characteristics of short video platforms. Next, the study provides a theoretical reference for improving short video platforms from the aspects of information, service, and system, which is conducive to the overall improvement of the short video platform. Finally, through the analysis of the above findings, the authors propose the following suggestions to enhance the willingness of users to share information in short marketing videos and achieve marketing objectives.

The platform environment should be optimized and the quality of marketing information should be improved. Information quality is closely related to user experience, and users pay more attention to information quality, where information content is an essential factor in evaluating the quality of short videos. Firstly, practitioners must pay attention to content and improve the quality of information. This is the most important step in attracting users. For example, they should always focus on matching product and marketing elements to deliver the right information to the product's audiences. Secondly, marketers need to be proficient in marketing content and be able to deliver the information to users in a comprehensive manner. This allows for a good flow of marketing information between marketers and users. Thirdly, short video platforms must review marketing information strictly, avoid exaggeration and false information, and control the source of information. Finally, it is necessary to push accurate information to users based on big data to maximize information efficiency.

Service quality should be improved and a bridge should be built between users and businesses. Service quality is crucial for users to perceive value, and an excellent quality of service brings the user closer to the business quickly. This is an important driver of proactive user behavior. Marketers and practitioners need to improve the quality of their services and give users a better consumer experience. Merchants should put the customer first and solve user problems in time. This includes pre-sale and post-sale issues with products. On this basis, merchants interact with users in real time to meet their individual demands, enhance user pleasure in the consumption process, and increase user trust in the platform.

Technological innovation should be continued and hardware and software adaptation should be improved. The rapid development of science and technology has accelerated the update iteration speed of mobile devices, including the system update of manufacturers' mobile devices. This frequently leads to adaptation problems between short video software and mobile devices. The adaptation problem seriously affects the system's quality and reduces the user experience. Therefore, the short video platform should continue to innovate in technology and repair the system problems of the platform in time. This will improve the adaptability of the software and hardware to ensure the smoothness of the software. As a result, it enhances the user's perceived control and operating experience and provides a high-quality system service for the dissemination of marketing information.

In short, it is important to put the user experience at the forefront. It is the overall perception of the user during browsing marketing information, receiving services from the platform and merchants, and operating the short video platform overall that has an important role in influencing individual behavior. Giving users a better sense of experience from the three aspects of information, service, and system is very important. As an online marketing method, short video marketing lacks the real experience of the consumption process. Therefore, marketers and practitioners should enhance the user experience during the process of browsing information, which can improve user satisfaction greatly and promote the occurrence of marketing information sharing.

## 6. Limitations and Further Research

There are still some limitations to this study. Firstly, from the perspective of information system success theory and flow experience theory, the current study investigated the influencing factors of users' willingness to share marketing information. However, the characteristics of the product were not explicitly considered. For example, a personalized product that only needs to be used by itself would also affect the user's willingness to share information. Secondly, owing to the fact that users are reluctant to let more people know about information involving personal privacy, the numbers of interviewers were likely to be limited. A further study could solve this problem and expand on the sample numbers of the research topic. Finally, the questionnaire data collection methods in this paper were limited to the WeChat platform and Credamo. Users' willingness to share marketing information on short video platforms may vary due to the functional groups. There is still room for improvement. Therefore, it is possible to increase the data acquisition, expand the data capacity, and make the research results more accurate in future related research.

**Author Contributions:** Conceptualization, R.S.; methodology, R.S., M.W. and C.L.; software, M.W.; investigation, N.G.; writing—original draft preparation, M.W.; writing—review and editing, R.S. and C.L. All authors have read and agreed to the published version of the manuscript.

**Funding:** This research work was supported by the Hebei Provincial Social Science Foundation Youth Project (HB19XW003), the Hebei Province Social Science Development Research Project Youth Project (2019041205003), the Humanities and Social Sciences Research Project of the Ministry of Education (20YJC860027), and Natural Science Foundation of Hebei Province Youth Science Fund (G2021203014), and the Master's Degree Students in the Innovative Ability Development Funding Project of Hebei Province (CXZZSS2023052).

**Institutional Review Board Statement:** Not applicable.

**Informed Consent Statement:** Not applicable.

**Data Availability Statement:** Data sharing is not applicable.

**Conflicts of Interest:** The authors declare no conflict of interest.

## Appendix A

**Table A1.** Interview Content.

| Variable Relationships | No | Item |
|---|---|---|
| Information systems success theory and perceptual control | 1 | How does information quality affect the user has perception control? |
| | 2 | How does service quality affect the user has perceived control? |
| | 3 | How does system quality affect the user has perceived control? |
| Information systems success theory and perceived pleasantness | 4 | How does information quality affect the user has perceived pleasantness? |
| | 5 | How does service quality affect the user has perceived pleasure? |
| | 6 | How does the system's quality affect the user has perceived pleasantness? |
| Personal experience and willingness to share | 7 | How does perceptual control affect the users' willingness to share marketing information? |
| | 8 | How does perceived pleasantness affect the users' willingness to share marketing information? |

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
