# Peer review of "The Influence of Short Video Platform Characteristics on Users’ Willingness to Share Marketing Information: Based on the SOR Model"

_sustainability, doi:10.3390/su15032448_

Round 1

Reviewer 1 Report

In this article, the author(s) look at the relationship between information system success factors, flow experience and willingness to share information. I enjoyed reading this article; however, I do think the author(s) need some additional details to better make their case for contributing to this body of knowledge. I think the article would be strong with some additional discussion. My suggestions by section are described below -

Major comments

Section 1

The discussion in this section is somewhat clear, however, additional discussion on gaps in the literature may strengthen the position of the paper. Specifically, the discussion on willingness to share information could be enhanced by further discussion.  

Section 2

The discussion on SOR theory is good. However, I do believe some of the recent literatures are missing and should be discussed in this section.

Further, the hypotheses should be developed based on the grounded theory. How does SOR theory support the hypotheses?     

The hypotheses indicate the direct relationships of constructs. Where is the hypothesis in relation to flow experience as a mediator? What is the theory behind the mediation?

Overall, this section needs much work than other sections and additional discussions on literature should be included. The literature discussed is quite old and should be replaced by new references (see Man-yee et al., 2012 for an example).

Section 3

The arguments in this section should be supported by recent literature. For example, the authors claim that “Qualitative and quantitative research methods have recently been widely used in the social sciences”. References are required here.

A major concerning issue is the number of participants in qualitative study. 6 participants may not give the valid results. How would say they are giving the valid results? I actually did not find any reason behind conducting the qualitative study.

What were the selection criteria in the quantitative study. in this study, 306 valid questionnaires were received. What criteria were used to check the validity?

There is no discussion on mediation in the analysis section although the authors mentioned mediation effect in the abstract section.

Section 5

This section is not adequate. Based on the findings, this study provides valuable insights on the variables examined. I believe the findings of this study should be of particular importance to marketers and practitioners. Therefore, additional implications should be included in this section.   

Minor comments

Some errors need to be corrected -

The manuscript should be checked by a professional editor. There are several displacements of punctuations and grammatical and spelling errors that need to be corrected. Some examples are - 

Abstract: The sentence starts with “And studies are…”, remove “and”.

Introduction Page 1: The sentence starts with “build a large following…”; please consider rewriting the sentence.

Some references are almost 10 years old or more and that need to be replaced by recent references.  

Author Response

Question 1 :    Section 1-The discussion in this section is somewhat clear, however,  additional b discussion on gaps in the literature may strengthen the position of the paper. Specifically, the discussion on willingness to share information could be enhanced by further discussion.  

Author’s Response:         Thank you for your suggestions! We have reorganized the discussion of willingness to share information in the introduction section. This can be seen on the second page of the paper in rows 54 to 83. Overall it has been refined in terms of both the consumers themselves and the consumption environment, and the national and international references have been recalculated on the basis of the refinements. This leads to the main elements of the paper on the impact of platform characteristics on the willingness to share marketing information.

Question 2:                       Section 2 - The discussion on SOR theory is good. However, I do believe  some of the recent literatures are missing and should be discussed in this section.

Author’s Response:       Thank you for your suggestions! As the reviewer suggested, the introduction to the concept of SOR theory and the current state of research is more than adequate and no additions have been made. Based on the first draft, we have sorted out and summarized the stimulus (S) and the organismic response (O) with the research. We have extensively explained the relationship of variables in a revised dissertation to elaborate their logical relationships with each other, factors affecting their behaviour, and outcomes. Examples include literature 25 and 28. This can be seen on page 3 of the paper, rows 126. This explains more clearly the meaning of the SOR theory and its key role in the paper, showing the relationship between the variables.

Question 3:                       Further, the hypotheses should be developed based on the grounded   theory. How does SOR theory support the hypotheses?

Author’s Response:   Thank you for your suggestions! The SOR theory states that various  aspects of the external environment act as stimuli that together influence the internal state of the individual, which in turn leads to responsive behaviour. Based on the reviewer's suggestions, We have redesigned the overall content of Section 2 and added new references. This can be seen on page 3 of the paper, row 118. We have used three platform characteristics from Information Systems Success Theory as stimulus factors that influence individuals. The individual perception of the flow experience as an organismic response indicates the user's reaction to the stimulus. And the elaboration of both theories has been newly enhancement. On this basis the relationship between the variables is analyzed in a new paragraph, leading to the hypothesis of the study.

Question 4:                       The hypotheses indicate the direct relationships of constructs. Where is  the hypothesis in relation to flow experience as a mediator? 

Author’s Response:        Thank you for your suggestions! We have added the assumption of mediating effects. This can be seen in row 254 to row 267 on page 6 of the paper. And we have redesigned the second section of the paper. In the area of flow experience first we have analyzed the content and the current state of research on flow experience. We found that in numerous studies the flow experience has a clear mediating role in the middle between external stimuli and individual responses. Secondly, we analyzed the impact of platform characteristics on users' flow experience and the impact of users' flow experience on their willingness to share information. This leads to the following hypothesis about the flow experience.

Question 5:              What is the theory behind the mediation?

Author’s Response:        Thank you for your suggestions! The main reason for using the theory of flow experience as a mediator in this paper is through the analysis and collation of previous research. A flow experience is a state in which an individual is completely immersed in an activity and enjoys the experience deeply, ignoring everything else. It is the individual's reaction to an external stimulus. This in turn affects the individual's behaviour. Numerous scholars have identified a clear mediating role for flow experiences between external stimuli and individual responses. This literature is also presented within this paper. This can be seen in row 255 on page 6 of the paper.

Question 6:                       Overall, this section needs much work than other sections and additional discussions on literature should be included. The literature discussed is quite old and should be replaced by new references (see Man-yee et al., 2012 for an example).

Author’s Response:         Thank you for your suggestions! We have made extensive adjustments to the second section of the paper. The reviewer raised the issue of outdated references and We have updated them and kept some of the classic literature. For example, We have updated the literature from Man yee and others to the paper published in 2021 by Zhang and others.

Question 7:                       Section 3-The arguments in this section should be supported by recent literature. For example, the authors claim that “Qualitative and quantitative research methods have recently been widely used in the social sciences”. References are required here.

Author’s Response:       Thank you for your suggestions! In many papers that use structural equations, the author obtains data in the form of a questionnaire and analyses it to draw conclusions. This is the more conventional method of writing. My reading of the literature revealed that qualitative analysis is also a good method of research. The combination of qualitative and quantitative is more likely to show the rigour and science of the research process. As suggested by the reviewer, We have also added references. Please reviewers look at the literature 54. The literature 55 has also been written using a shaped mixed research approach. Also the literature 56 provides a reference for the use of mixed research methods. This can be seen on page 6 of the paper, row 275.

Question 8:                       A major concerning issue is the number of participants in qualitative study. 6 participants may not give the valid results. 

Author’s Response:        Thank you for your suggestions! Qualitative research can clarify the  specific structure of each theory in reality and is a safeguard for the research in this paper. As suggested by the reviewers, we consider six interviewees to be a small number indeed.  After referring to the relevant papers used in the mixed methods, we decided again on the number of interviewees. We have consequently re-contacted school personnel to increase the number of interviewees to 12. Although the number of interviewees did not reach several dozens, there was good coverage in terms of age and occupation. The sample may be considered to be somewhat representative. In addition, the interview method used in this paper is mainly to explore the surface relationships between variables. The specific analysis is still to be achieved through structural equation modelling.

Question 9:              How would say they are giving the valid results?

Author’s Response:        Thank you for your suggestions! We designed the content of the interviews by reading the literature. The content of the interviews allows us to understand what individuals really think about the variables in this study and to ensure the scientific validity of the study. The selected interviewees were all proficient in operating smartphones and short video platform software, and had experience in consuming short video platforms. As a result, they have unique ideas about the characteristics of the platform and their own experiences. By designing scientifically based interview questions we can capture the relationship between real variables.

Question 10:           I actually did not find any reason behind conducting the qualitative study.

Author’s Response:   Thank you for your suggestions! The description begins on page 6 of the      

paper at row 275. Also added literature 54 and 55 as a basis for using mixed research methods. Qualitative and quantitative methods are interlinked and complementary in research. Qualitative methods enable the researcher to clarify the specific structure of each theory in reality based directly on the analysis of the interviewees' performance, and to have an all-round control of the overall research structure. The combination of qualitative and quantitative methods of analysis allows for more accurate conclusions to be drawn.

Question 11:            What were the selection criteria in the quantitative study. 

Author’s Response:       Thank you for your suggestions! We have been adding selection criteria from quantitative studies to the paper. It can be seen on page 8 of the paper, row 372. The study was conducted with consumers who had experience of shopping on short-form video platforms. Questionnaires were distributed through the professional questionnaire websites Questionnaire Star and Credamo, and non-compliant respondents were excluded from the questionnaire collection and analysis process. The main categories of non-compliant interviewees are the following three types of people. The first is the people who doesn’t have shopping experience from short video platforms. The second is the people who has highly consistent questionnaire results or no differentiation. The third is the people who gives contradictory answers or reacts to data falsely.

Question 12:                     in this study, 306 valid questionnaires were received. What criteria were used to check the validity?

Author’s Response:         Thank you for your suggestions! It is elaborated from page 9 of the paper, row 406. We used SPSS 26.0 statistical software and AMOS 22.0 software to analyse the reliability and validity of the scales. Firstly, the internal consistency reliability of each factor is analyzed by calculating the Cronbach's α coefficient. All values should be greater than the standard value of 0.7. In addition, the AVE value of each latent variable is more significant than 0.5, and the composite reliability CR is more significant than 0.7, indicating that the observation indicators have good consistency and that the convergent validity of the data is ideal. Finally,The square root of the AVE value of each variable is significantly larger than the correlation coefficient between the observed variables. The scale has good discriminant validity.

Question 13:                     There is no discussion on mediation in the analysis section although the  authors mentioned mediation effect in.  

Author’s Response:   Thank you for your suggestions! We have added on the discussion of mediating effects. The discussion begins on page 14, row 569 of the paper. This study was conducted using the Bootstrap method for mediating effect analysis. Perceived control and perceived pleasure partially mediated the relationship between the characteristics of the short video platform and users' willingness to share marketing messages. And the mediating effect of perceived pleasure in this is stronger than perceived control.

Question 14:       Section 5-This section is not adequate. Based on the findings, this study  provides valuable insights on the variables examined. I believe the findings of this study should be of particular importance to marketers and practitioners. Therefore, additional implications should be included in this section.

Author’s Response:     Thank you for your suggestions! As the reviewer suggested, the results  of this study should be of particular importance to marketers and practitioners. So we have added in section 5 of the paper. It can be seen in row 592 on page 14 of the paper. The significance of this study is to improve the dissemination of marketing messages on short video platforms, which in turn promotes marketing effectiveness. Based on the first draft, we have refined three aspects of the short video platform: the quality of the message, the quality of the service and the quality of the system. In addition, the importance of user experience in the process of information sharing is emphasized. We have added this part in the last paragraph of this section. The characteristics of short video platforms affect user experience, which in turn affects users' willingness to share marketing information. Short video platforms and businesses must pay attention to user experience while improving the quality of information, services and systems.

Question 15:      Minor comments - The manuscript should be checked by a professional  editor. There are several displacements of punctuations and grammatical and spelling errors that need to be corrected. Some examples are - Abstract: The sentence starts with “And studies are…”, remove “and”.

Author’s Response:     Thank you for your suggestions! We have made changes and corrections  to the full text.

Question 16:      Introduction Page 1: The sentence starts with“build a large following…”;  please consider rewriting the sentence.

Author’s Response:  Thank you for your suggestions! We have amended the sentence.

Question 17:   Some references are almost 10 years old or more and that need to be replaced by recent references.

Author’s Response:   Thank you for your suggestions! In conjunction with the research we  have added some new literature, which we have used to improve the scientific nature of the research. For the more outdated literature, we have updated it through a review of the literature.

Reviewer 2 Report

In my opinion, the Authors should consider broadening the research implications part as well as the limitations. The sample size was relatively small - and the Authors don't mention that in the limitations. This part should also be redesigned. 

Author Response

Thank you for your suggestions! we have added some points in section 5 of the paper. It can be seen in row 592 on page 14 of the paper. The significance of this study is to improve the dissemination of marketing messages on short video platforms, which in turn promotes marketing effectiveness. Based on the first draft, we have refined three aspects of the short video platform: the quality of the message, the quality of the service and the quality of the system. In addition, the importance of user experience in the process of information sharing is emphasized. We have added this part in the last paragraph of this section. The characteristics of short video platforms affect user experience, which in turn affects users' willingness to share marketing information. Short video platforms and businesses must pay attention to user experience while improving the quality of information, services and systems.

For the limitations of the study we have added one point to the first draft. That is, people are reluctant to share information related to personal privacy. In addition, although the number of interviewers was increased to 12, the number of interviewers and questionnaires in the quantitative analysis was likely to be limited. Further study could expend the sample numbers of the research topic.

The number of interviewers in the first draft was 6. As suggested by the reviewer, We re-contacted the School People staff to add to the number of interviewers, increasing the number to 12. Based on the reading of previous literature and the practicalities of this paper's qualitative research, 12 people is a more reasonable number. The quantitative analysis had 306 valid questionnaires to prove the research hypothesis and met the sample size requirements of the structural equation model.

We have added the sample size issue into the section of limitations.

Reviewer 3 Report

The article brings into attention a topic of interest regarding intention to share marketing information manifested by the users of short video platform. In order to fulfill the research purpose, a qualitative and a quantitative research were employed. For improving the quality of the paper, I have some recommendations:

In the Introduction, the idea presented in rows 57-59 is not clear.

In Research and Data analysis section:

-       The items used to define the latent variables should be displayed

-       At row 319 I think instead of combined reliability you referred to composite reliability.

-       In mediating analysis the Variance Accounted For (VAR) value should be calculated and establish the type of mediation (full of partial mediation). 

Author Response

 In the Introduction, the idea presented in rows 57-59 is not clear.

Author’s Response:        Thank you for your suggestions! Rows 57-59 focus on the factors that influence users' willingness to share as analyzed by current scholars from different perspectives. This sets the scene for the narrative that follows. As requested by the reviewer, we first state that the study of factors influencing the sharing of marketing messages is a relatively hot topic. It is further stated that the current influencing factors are mainly both the consumers themselves and the consumer environment. We have redesigned the section, reordering the literature accordingly. Literature 14 and 15 have been added to supplement the content.

 In Research and Data analysis section: The items used to define the    latent variables should be displayed.

At row 319 I think instead of combined reliability you referred to composite reliability.

In mediating analysis the Variance Accounted For  value should be calculated and establish the type of mediation (full of partial mediation).

Author’s Response:          Thank you for your suggestions! In the article we have applied latent variables for writing and analysis. In row 319, due to a spelling issue we wrote composite reliability instead of combined reliability, this has been corrected. The revision can be found on page 10 of the paper, row 444. This study was conducted using the Bootstrap method for mediating effect analysis. All paths do not include 0 between the upper and lower limits of the 95% confidence level. Perceived control and perceived pleasure partially mediated the relationship between the short video platform Characteristics and users' willingness to share marketing messages.